# Incorporation of Fly Ash in Flame-Retardant Systems of Biopolyesters

**DOI:** 10.3390/polym15132771

**Published:** 2023-06-21

**Authors:** Marcos Batistella, Jean-Claude Roux, Nour-Alhoda Masarra, Gwenn le Saout, Constantinos Xenopoulos, José-Marie Lopez-Cuesta

**Affiliations:** 1Polymers Composites and Hybrids, IMT Mines Alès, 30319 Ales, France; jean-claude.roux@mines-ales.fr (J.-C.R.); nour-alhoda.masarra@mines-ales.fr (N.-A.M.); jose-marie.lopez-cuesta@mines-ales.fr (J.-M.L.-C.); 2LMGC, IMT Mines Ales, Université Montpellier, CNRS, 30100 Ales, France; gwenn.le-saout@mines-ales.fr; 3Holcim Innovation Center, 38090 Saint Quentin Fallavier, France; constantinos.xenopoulos@holcim.com

**Keywords:** polybutyl succinate, polybutyl adipate terephthalate, fly ash, fire behavior, surface modification

## Abstract

The incorporation of fly ash in polybutyl succinate (PBS) and polybutyl adipate terephtalate (PBAT) in the partial replacement of ammonium polyphosphate and/or melamine polyphosphate is evaluated in the present work. Furthermore, the influence of the surface modification of fly ash with two silanes and titanate coupling agents was also studied. Cone calorimeter experiments, pyrolysis combustion flow calorimeters (PCFCs), and UL94V tests were used to assess the fire performance of the composites. Scanning electronic microscopy, X-microanalysis, and X-ray diffractometry analysis were carried out on cone calorimeter residues in order to access the fire-retardant mode of action. The formation of new components due to the presence of fly ash was highlighted by X-ray diffractometry, indicating the synergistic effects between the flame-retardant system and fly ash. The X-microanalysis results showed that the main fraction of initial phosphorous is present in the cone calorimeter residue, indicating that the proposed system acts in a condensed phase.

## 1. Introduction

Fly ash is a waste byproduct issued from thermal power plants, and around 750 million tons are generated each year in the world, leading to a serious concern about their disposal. However, they have not been used fully regarding their true potential as a vital resource of engineering materials [1]. The pulverization of coal in power plants entails the formation of diverse shapes and sizes of fly ash particles. Some of them are spherical and called cenospheres, according to the Greek words kenos (hollow) and sphaira (sphere). They correspond to a spherical solid with a hollow center, and are one of the most important and interesting fractions of fly ash, owing to their mechanical resistance density, high thermal insulation and stability, and hydrophilic nature, which allow for wide range of industrial applications, particularly in cementitious composites [2,3,4,5,6]. The composition of fly ash and, particularly cenospheres, is mainly based on 30 to 90 wt% of amorphous glassy phases of aluminosilicates and various crystalline phases [7,8]. The other components are mullite, quartz, and feldspar [8]. Their dimensional range is very wide, from submicronic particles around 0.5 µm to more than 100 µm [9].

Fly ash has been considered a very interesting material for reinforcing fillers in polymers to obtain eco-friendly composites [10,11], and the word biocomposites has been used for reinforced thermoplastics with fly ash [12]. Recycled mineral waste or by-products of fly ash, particularly those containing cenospheres, have two advantages: their relatively low cost in comparison with engineered reinforcement materials and their low density. Consequently, they can reduce the structural weight of composites used in transportation without the loss of mechanical properties. Many studies report the use of fly ash in various polymers such as particularly polyolefins [13,14,15], PVC [16,17], ABS [18,19], epoxy [20,21], and polyester resins [22,23].

A recent paper [24] deals with the incorporation of fly ash cenospheres into polylactide (PLA). After studying the mechanical properties of the filled polymers, the authors investigated the influence of UV radiations on the evolution of crystallinity and physicomechanical properties.

On the whole, the main properties investigated for fly ash/polymer composites have been static and dynamic mechanical, tribological [25], and dielectric properties [26]. The influence of the use of silane or titanate coupling agents on mechanical properties was investigated in various studies, particularly for fly ash in butadiene rubber [27].

To our knowledge, interest in using fly ash to modify the fire behavior of polymers has only been shown by [28]. These authors combined fly ash with mixtures of ammonium polyphosphate, pentaerythritol, melamine, and boric acid for water-based intumescent coatings applied on structural steel. They noticed that fly ash could improve the quality of the intumescent char formed and also impart more stability regarding oxidation. X-ray diffractometry produces evidence of chemical interactions between fly ash with the coating ingredients at high temperatures.

In this study, we have investigated various compositions of biopolyesters containing SuperPozz fly ash (South Africa), which is a fine and light-colored powder produced by the Holcim company. SuperPozz is mainly composed of amorphous calcium silicates and aluminates.

The original character of this work consists in assessing the performance of SuperPozz as a component of fire-retardant systems to improve the fire behavior of two biobased polymers: polybutylene succinate (PBS) and polybutylene adipate terephthalate (PBAT). Biopolymers appear promising for replacing oil-based plastics for durable applications. Both are biodegradable and are based on monomers produced today from biobased resources [29].

Our objective is to highlight possible synergistic effects on the fire reaction parameters or at least the prospect of substituting a fraction of flame retardant (FR) without any loss of fire performance.

This approach makes possible the reduction in the environmental footprint of flame retardant biopolyesters by reducing or replacing a fraction of synthetic FR with a mineral waste byproduct. For some compositions, the influence of surface treatments of the fly ash will be investigated. The development of biobased/biodegradable polymers is the focus of increasing interest from applications in various areas such as transportation, housing, or electrical appliances where durability is important and fire performance must be assured. The development of FR systems devoted to biopolyesters has been initially and mainly focused on PLA, owing to its predominant production among biopolyesters. However, since the use of PLA-based biopolymer blends is increasingly faced with inherent PLA-based drawbacks (ductility, crystallization issues), the attention begins also to be focused on the fire behavior of other kinds of biopolyesters.

To date, the majority of FR systems available for use with PLA are based on intumescent compositions, which are mainly based on ammonium polyphosphate (APP) and are often combined with biobased char promoters, such as lignin or other types of co-synergists [30,31]. So, as an initial approach, similar FR systems could be considered to improve the fire retardancy of PBS and PBAT. Nevertheless, whereas various research works have been performed on intumescent systems devoted to PBS, as reported by Xiao et al. [32], there is a marked lack of data on flame-retardant PBAT.

Concerning the types of intumescent FR systems used for PBS, some of them are entirely from biobased resources, including natural fibers, which have been modified or reacted with phosphorus compounds [33,34,35]. Chen et al. [36] used guanosine combined with phytic acid, a natural compound present in the bark of trees, which contains phosphorus.

Nevertheless, for PLA, the majority of intumescent flame-retardant systems (IFR) used in PBS are based on APP. Some authors [37] used only APP treated with a silane coupling agent and blended with PBS using an internal mixer. Some compositions were crosslinked by water. Significant improvements were noted for 15 wt% of APP (Limiting Oxygen Index (LOI) values from 24 to 28%) between crosslinked compositions and reference ones without silane and crosslinking.

In other research studies, APP was combined with inorganic compounds, including natural and synthetic nanoparticles.

Fumed silica was combined with an IFR system based on APP and melamine. For 2 wt% silica, 15 wt% APP, and 3 wt% melamine, an LOI value of 36% and a V-0 rating for the UL 94 vertical test were achieved.

In addition to silica, a synthetic layered double hydroxide (3 wt%) was combined with an intumescent FR system composed of APP and melamine (17 wt%) [38]. A LOI value of 33% and a V-0 rating for the UL 94 vertical test were achieved. Under an irradiance of 25 kW/m^2^, a value of 250 kW/m^2^ for the peak of heat released (pHRR) was obtained, in comparison with 368 kW/m^2^ for the virgin polymer.

Another work [39] evaluated combinations of natural layered silicates existing at a nanometric scale (halloysite and sepiolite) at 5 wt% with APP (for a global loading of 20 wt%), which were tested by cone calorimeter for an irradiance of 50 kW/m^2^. A slight interaction was observed in comparison with 20 wt% APP used on its own in PBS. Improved performance was achieved with a composition of 5 wt% alkali lignin, 5 wt% sepiolite, and only 10 wt% APP. The pHRR was significantly lowered to 267 kW/m^2^ (367 kW/m^2^ for 20 wt% APP). This was ascribed both to the role of lignin as a char promoter as well as the formation of new crystalline structures, such as NaMg(PO_3_)_3_, which were able to reinforce the intumescent and charred structure formed. Moreover, the formation of these new compounds ensured the conservation of the initial phosphorus content in the condensed phase.

APP and halloysite association were also considered, but with the use of soy protein instead of lignin. Compositions with 20 wt% APP, 10 wt% soy protein, and 1 wt% halloysite produced LOI values of 38% along with a V-0 rating. Moreover, cone calorimeter testing performed at 35 kW/m^2^ allowed the pHRR to be reduced from 538 kW/m^2^ for virgin PBS to 371 kW/m^2^, and the total heat released was from 64.7 to 42.4 MJ/m^2^.

In the present study, various binary or ternary compositions of the above biopolyesters containing fly ash, possibly surface-treated using two silanes and a titanate, are processed at constant loading and assessed using the cone calorimeter and UL 94 tests. APP, MPP (melamine polyphosphate), and their blends are selected as FRs. The use of increasing amounts of fly ash will indicate the acceptable upper limit, leading to no detrimental influence on fire performance. The study of the degradation pathway of the biopolyester compositions through relevant characterization techniques is expected to provide significant information on the possible chemical reactivity of fly ash and its potential as an FR co-synergist.

## 2. Materials and Methods

### 2.1. Materials

Two biopolyesters were used in order to evaluate the influence of fly ash: a polybutyl succinate (PBE003 from NaturePlast, Mondeville, France) and a polybutyl adipate terephthalate ecoflex™ Ecoblend C1200 from BASF (Ludwigshafen, Germany).

Fly ash (SuperPozz supplied by Holcim) noted SP with a specific surface area of 1.55 m^2^/g, a median diameter of 6 µm, and a cut-off diameter of 100 µm were produced by Holcim. A detailed composition was obtained by X-ray fluorescence and X-ray diffraction Rietveld analysis refinement of powder XRD data (Appendix A and Appendix A).

Three coupling agents were employed according to the type of polymer used. Two silanes, 3-Aminopropyl)triethoxysilane (noted A), 3-Glycidyloxypropyl)trimethoxysilane (noted G) from Sigma (St. Louis, MO, USA) and a titanate, isopropyl tri[di(octyl)phosphato] titanate (KR12) (noted T) from Kenrich Chemicals (Bayonne, NJ, USA), were used. Ethanol (96% *v*/*v*-PanReac, Darmstadt, Germany) was also used.

Two flame retardants, ammonium polyphosphate (Exolit AP422, from Clariant, Hürth-Knapsack, Germany, noted AP) and melamine polyphosphate (MP200, from BTC, Monheim, Germany, noted MP), were used. SuperPozz fly ash was supplied by Holcim.

### 2.2. Methods

#### 2.2.1. Processing of Compositions

Formulations were prepared using a twin-screw extruder (Clextral, Firminy, France) with a feed rate of 5 kg/h. The temperature profile of each polymer is shown in Table 1. The nomenclature of binary and ternary compositions at 25 wt% global loadings used throughout are shown in Table 2 and Table 3.

#### 2.2.2. Surface Modifications

Surface treatments were carried out as follows. A total of 50 g of SuperPozz was dispersed in 500 mL of an ethanol/water mixture (96/4). By means of continuous stirring, the pH of the solution was modified while stirring with the addition of acetic acid to a target pH of 4.5. A coupling agent was added next with a concentration of 1 wt%, relative to the amount of SuperPozz. These mixtures were stirred for 6 h at 80 °C. At the end of the reaction, the solvent was evaporated and the SuperPozz was washed three times with ethanol to remove excess reagent.

#### 2.2.3. Thermal Stability and Fire Reaction

The thermal stability of composition was evaluated using Setaram equipment (SETSYS Evolution, Caluire-et-Cuire-France) under nitrogen at 40 mL/min, from 30 to 750 °C with a temperature ramp of 10 °C/min. Square specimens of 100 × 100 × 4 mm^3^ for cone calorimeter tests were prepared using Krauss Maffei (Germany) injection molding equipment.

The Pyrolysis Combustion Flow Calorimeter (PCFC) (Fire Testing Technology, West Sussex, UK) aims to determine fire reaction parameters at a microscopic scale according to Lyon [40]. Around 3 mg of polymer material was pyrolyzed in N_2_ at 1 °C/s from ambient temperature to 750 °C. Then, the gaseous thermal degradation products were burnt off in a furnace (combustor) with excess oxygen at 900 °C to achieve complete combustion. The heat release rate (HRR) was measured as a function of temperature, leading to the determination of the peak value of the HRR, corresponding time, and the total heat release (THR) and heat release capacity (HRC).

Cone calorimeter tests were carried out using FTT equipment according to ISO 5660-1 with an irradiance of 50 kW/m^2^. Three specimens were tested and mean values of Time to Ignition (TTI), the peak of heat release rate (pHRR), total heat released (THR), Maximum Average Rate of Heat Emission (MARHE), and mass loss were recorded, as well as the percentage of remaining residue.

The ability to self-extinguish for the different compositions was evaluated using the UL94V test on 90 × 10 × 4 mm^3^ samples fabricated with the above-mentioned injection molding equipment.

#### 2.2.4. Study of Microstructures and Composition of Residues

A Scanning Electron Microscopy (SEM) Quanta 200 FEG (FEI Company, Hillsboro, OR, USA) operating in a high vacuum at an acceleration voltage of 12.5 kV was used to investigate the microstructure of the combustion residues after cone calorimeter tests. Energy dispersive X-ray spectroscopy was also carried out on residues using an Oxford XmaxN system and a detector with a resolution of 133 eV. In order to assess the formation of new compounds resulting from possible interaction between flame retardants and fly ash, X-ray diffraction of the cone calorimeter residues was performed, using a D8 Advance diffractometer (Bruker AXS with CuKα radiation and a Lynxeye detector, Billerica, MA, USA) in the 10–60° 2θ range with a steep size of 0.007°.

## 3. Results

### 3.1. Thermal Analysis

The thermal decomposition of PBAT occurs in one step from 400 °C to 450 °C (Figure 1). A residue of 7% stable up to 750 °C is obtained. This can be explained by the role of the aromatic structure of the terephthalic acid which leads to decomposition to a polyaromatic structure similar to the polyethylene terephthalate (PET). The combination of PBAT with only APP (PBAT-1) exhibits a lower thermal stability than pure PBAT due to the presence of APP, which decomposes from 350 °C and can entail a phosphorylation of PBAT. The occurrence of this reaction is supported by the comparison between the experimental TGA curve of PBAT-1 and the theoretical thermogram built from the respective TGA curves of APP and PBAT. It can be shown that the kinetics of mass loss is the fastest for the experimental curve. Moreover, the residue is lower, indicating reactivity between the two components.

All the other compositions with flame retardants exhibit lower thermal stability than pure PBAT due to the presence of APP (Figure 2). The lower values of onset degradation temperature, which are noted for the compositions with both FRs and surface-treated fly ash, are ascribed to the thermal decomposition of the surface coupling agent below 350 °C and the release of melamine from melamine polyphosphate.

All these compositions show higher values of residues than PBAT alone (Table 4) due to the presence of flame retardants acting in the condensed phase, as well as the presence of SuperPozz, which is very thermally stable (only 2 wt% of mass loss at 750 °C). Consequently, it could be expected that the residue will be highest with a higher percentage of SuperPozz. Indeed, compositions with 12.5 wt% of fly ash lead to the highest residues, much higher than with APP alone. Nevertheless, most parts of the residues of compositions with melamine polyphosphate are lower than the value obtained with APP. This can be ascribed to the release of melamine and the formation of new compounds with limited thermal stability beyond 450 °C.

In contrast to PBAT, PBS undergoes total decomposition in one step and a similar temperature range. Nevertheless, the comparison between the theoretical and experimental curves of PBS-1 shows a strong interaction between APP and PBS during their thermal decomposition (Figure 3). This also can be explained by phosphorylation of PBS and possibly also by hydrolytic processes related to the release of water during APP decomposition. In this case, the difference for the temperature of maximum degradation mass loss is close to around 50 °C, whereas it is only around 20 °C for PBAT. The amount of residue is nearly the same and very close to PBAT-1 (20%).

As noticed in previous studies (for example, [39]), the presence of FRs based on APP lower significantly all the characteristic degradation temperatures reported in Figure 4 and seen in Table 5. However, the composition of the FR system does not influence these temperatures. The quantities of the final residues are between 20 and 24%, except for the composition with both FRs and without fly ash. This can be ascribed to the melamine release from MPP. It is interesting to note that the compositions with the highest SuperPozz amount do not lead to the highest residues (similar values as APP alone). Higher values are obtained for the four compositions with modified SuperPozz. This seems to indicate that more stable new compounds are formed between polyphosphates and some compounds of SuperPozz, in comparison to what is observed for the corresponding compositions of PBAT.

### 3.2. Pyrolysis Combustion Flow Calorimeter

PCFC data for PBAT are presented in Table 6. It can be noted that all compositions lead to a decrease in the pHRR, and the time corresponding to the peak value also decreases. This can be ascribed to the decomposition of APP and its reaction with the polymer. Only compositions with APP mixed with surface-treated SuperPozz at 6.25 wt% allow the pHRR value to be reduced in comparison with compositions with APP alone. Conversely, mixed compositions with melamine polyphosphate bring about a strong increase in the pHRR and HRC at a similar level as PBAT alone. This is consistent with the TGA results and shows that the thermal degradation of these compositions leads to a residue that is less stable than APP and SuperPozz combinations.

The outstanding behavior of PBS compositions tested by PCFC can be seen through the strong difference between the profiles for pure PBS and flame-retardant PBS with APP (Table 7). As noted on the TGA curves, APP dramatically reduces the thermal stability of this biopolyester. The effect obtained is more pronounced than with PBAT (where the T20% for PBS is reduced by 41 °C compared to 16 °C for PBAT). This can be attributed to the aromatic structure of PBAT, which allows a lower degree of phosphorylation by APP and possibly a better resistance to hydrolysis resulting from the release of water from APP decomposition. Hence, the PCFC curve of PBS-1 is very sharp, with a higher pHRR value than PBS, whereas its THR value is more significantly reduced than the corresponding PBAT composition. Conversely, the combination of APP and MPP (PBS-8) leads to a strong decrease in the pHRR and a significant residue. This shows the benefit of the presence of melamine, which is able to produce endothermal effects during its decomposition and reduce the exothermal effect from the burning of short PBS chains due to the influence of APP.

For all compositions with SuperPozz, the pHRR is reduced in comparison with PBS-1, but all curves are sharper than PBS alone and PBS-8. The mixed compositions with 12.5 wt% SuperPozz with only APP tend to exhibit lower pHRR values as well as higher amounts of residues. The use of surface treatments on SuperPozz produces significant differences in pHRR values for the same compositions, but no conclusion can be drawn about the influence of each type of coupling agent. Nevertheless, it can be noted that similarly to PBAT, compositions with both flame retardants and SuperPozz lead to high pHRR values and low residues, showing the limited stability of the chemical structures formed in the condensed phase.

### 3.3. Cone Calorimeter Tests

The HRR curve of virgin PBAT (Appendix A) is very sharp and corresponds to a very flammable material that burns very quickly. The use of APP at 25 wt% allows the pHRR to be reduced by 36% and extends the duration of the combustion (Figure 5 and Figure 6 and Table 8). The combination of APP with MPP seems slightly more advantageous, but the presence of two successive peaks indicates that no protective layer has been formed at the surface of the material. Combinations between APP and the untreated and surface-treated SuperPozz at 6.25 wt% allow HRR values to be reduced, particularly for the compositions with the epoxy coupling agent. The TTI is delayed, and the duration of the combustion is enhanced, leading to a lowering of the MARHE. Moreover, equimassic compositions of SuperPozz and APP also lead to superior fire performance compared to APP alone, with the best results for the epoxy coupling agent. So, a synergistic effect can be observed for the pHRR and MARHE parameters up to a substitution of 50% APP by SuperPozz.

The combined use of both FRs with SuperPozz leads to outstanding behavior. HRR curves present two or three peaks, the last one being higher, and occur beyond 500 s. Consequently, the mass loss is significantly delayed (Figure 6) and the MAHRE is dramatically reduced. Moreover, the type of coating agent for these compositions strongly influences performance. In fact, the titanate coupling agent makes it possible to further delay and reduce the pHRR as well as the mass loss in comparison with the other mixed compositions. Additionally, the first peak in the case of untreated SuperPozz disappears for the compositions with the modified material. Consequently, it can be suggested that a very cohesive protective layer is built using compositions with 12.5 wt% APP, 6.25 wt% MPP, and 6.25 wt% of surface-treated SuperPozz. UL94 results show that the addition of APP leads to V-2 classification. The partial replacement of APP by FA or MP does not lead to an improvement in the observed behavior.

Cone calorimeter curves of PBS are presented in Figure 7 and Figure 8 and complementary results are supplied in Table 9. The HRR curve of virgin PBS is less sharp than PBAT. In addition, for all the flame-retardant compositions, the pHRR appears very early, around 100 s, which is similar to all the corresponding compositions of PBAT. This could be ascribed to the influence of APP on the biobased macromolecules. As seen in TGA thermograms, APP strongly impairs the thermal stability of PBS before promoting a charred structure. The use of MPP and SuperPozz cannot offset this effect. Nevertheless, once again, compositions with 12.5 wt% APP, 6.25 wt% MPP, and 6.25 wt% of surface-treated SuperPozz impart a synergistic effect on the fire reaction parameters and delay the mass loss (Figure 8). It can also be remarked that titanate coupling is more beneficial, albeit the performance is less outstanding in comparison with PBAT. The benefits of employing coupling agents can be viewed through the decrease in the MAHRE, allowing the HRR peaks to be shifted to higher time values. Their action could be explained by a better dispersion of the fly ash leading to a better interaction with flame retardants, as well as a superior coupling with the polymers and a concomitant improvement of the cohesion of the residue. UL94 results show that the addition of APP leads to V-2 classification. The partial replacement of APP by FA does not lead to an improvement in the observed behavior. Interestingly, the addition of MP leads to a V-0 classification, probably due to an improved char formation which is able to limit the flame propagation.

### 3.4. SEM Observations and an X-Microanalysis of Cone Calorimeter Residues

The observation of the residues using SEM corresponds to the upper part of the residues as well as their cross-section. Similar features have been identified for the two polymers when considering comparable compositions.

SEM micrographs of residues without SuperPozz are shown in Figure 9. For both polymers and flame-retardant system compositions, all the residue appears microporous. It can just be noticed that the microporous character and the fineness of the structure seem higher for compositions with MPP and PBS. Moreover, the cross-section of PBS containing both FRs appears very compact.

For all the compositions containing SuperPozz, the residues exhibit crystalline structures appearing as platelets, which are often organized as windroses (Figure 10). These structures are visible at the surface of residues but can also be found in the cross-section.

In some cases, spheroidal structures made of platelets can be identified as well as initial fly ash particles, with some of them exhibiting the early stages of transformation into new structures (Figure 11). On the whole, there are no significant differences between residues from PBS or PBAT compositions, either as a result of the presence of MPP or the surface treatments of the fly ash.

The X-ray microanalysis of the residues is presented in Table 10 and Table 11. The phosphorus balance corresponds to the quantity of phosphorus remaining in the residue in comparison with the initial amount in the sample. As expected, the compositions without SuperPozz exhibit a strong weight percentage of carbon and also a significant percentage of phosphorus. For all the compositions with SuperPozz, the percentage of carbon is dramatically reduced, whereas the percentage of phosphorus is similar or even higher. This indicates stronger retention of carbon in the residue through the formation of the new chemical structures visible in the SEM micrographs. This effect is particularly notable for the ternary compositions in PBAT (PBAT-9-10-11) and seems apt to explain the high number of final residues and the global fire performance observed.

Figure 12 indicates the composition of the specific crystalline microstructures found in the residues visible by SEM. Two distinct compositions can be identified for windroses: one with a high percentage of phosphorus and some aluminum and the other with a very high P percentage accompanied by a noticeable percentage of calcium instead. The latter element is also significantly present in the spherical aggregate of platelets.

### 3.5. X-Ray Diffractometry of the Residues

Complementary investigations are needed to fully understand the mode of action involved as well as the structure of the protective layers. All the residues have been analyzed using XRD. Figure 13 and Figure 14 present the corresponding diffractograms for the raw fly ash and the following compositions of residues, respectively, for PBAT and PBS. A hump appears on the diffractograms of compositions without fly ash (presented here only for PBS-1), indicating the formation of X-ray amorphous structures. For the patterns corresponding to samples containing SP in their formulation, one notes, in addition to the amorphous hump, the presence of crystalline phases present in the raw fly ash and the formation of two components resulting from reactions between fly ash and flame retardants: an aluminum phosphite Al(PO_3_)_3_ and a hydrated ammonium phosphate (Al_3_(NH_4_)H_14_(PO_4_)_8_.4H_2_O). They are not dependent on the composition of the FR system. Moreover, the additional presence of hydrated iron phosphate seems likely. To our knowledge, these components were never reported in the composition of residues related to combinations between APP or APP + MPP combined with silicates in FR systems for polymers. As reported in the introduction, a study [28] combined fly ash with APP and pentaerythritol in ethylene vinyl acetate coatings for steel structures. The analysis of combustion residues by X-ray diffraction after exposure to a butane burner revealed the presence of SiP_2_O_7_ silicophosphates. In a previous article recently submitted [41], we carried out similar combinations between fly ash and the same flame retardants in polylactide. It was noted that this last polymer exhibited a lower degree of interaction with APP during its thermal decomposition in comparison with PBAT and PBS. The analysis of cone calorimeter residues also highlighted the formation of cohesive and protective layers corresponding to a strong retention of the initial phosphorus in the condensed phase, but the crystalline phases identified by X-ray diffraction were two ammonium phosphates, NH_4_AlP_2_O_7_ and Al(NH_4_)HP_3_O_10_. Therefore, the combinations performed in the present paper have led to the formation of specific compounds due to, on the one hand, the reaction of APP with the polymers and, on the other hand, the reactions of the flame retardants with fly ash.

## 4. Conclusions

Fly ash has been combined with ammonium polyphosphate and blends of ammonium polyphosphate and melamine polyphosphate in order to improve the flame retardancy of PBS and PBAT. Comparisons were also made with compositions containing only the flame retardants at the same global loading of 25 wt%. For some compositions, the fly ash was surface-modified using silane and titanate coupling agents. A strong reactivity of the phosphorous flame retardants toward both polymers is remarked, leading to lower thermal stability for the composites. Based on PCFC results, it appears that the combination of the two flame retardants with fly ash led to residues in both polymers with limited thermal stability for the two polymers. Nevertheless, cohesive residues are formed during the cone calorimeter test for all mixed compositions. Moreover, some interactions on the pHRR and MAHRE, in particular, are noted for the ternary compositions. For PBAT, the use of surface treatments seems advantageous insofar as, based on the evidence from HRR and mass loss profiles being shifted to higher time values, protective structures are formed during combustion that play a role in increasing cohesion.

The formation of new chemical structures from fly ash components that are observed by SEM observations, X-ray microanalysis, and X-ray diffraction seem responsible for the cohesive character of the protective layers formed. In the case of PBAT, the intrinsic charring potential of the macromolecule due to the aromatic structure of the terephthalic acid leads to better efficiency of these new compounds in combination with the residual char formed.

## Figures and Tables

**Figure 1 polymers-15-02771-f001:**
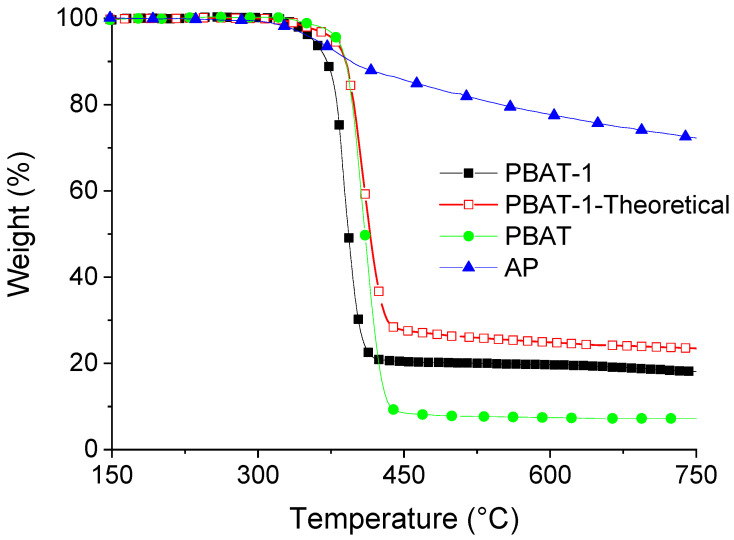
TGA curves (N_2_, 10 °C/min) of pure PBAT, AP, and PBAT-1 composition (experimental and theoretical).

**Figure 2 polymers-15-02771-f002:**
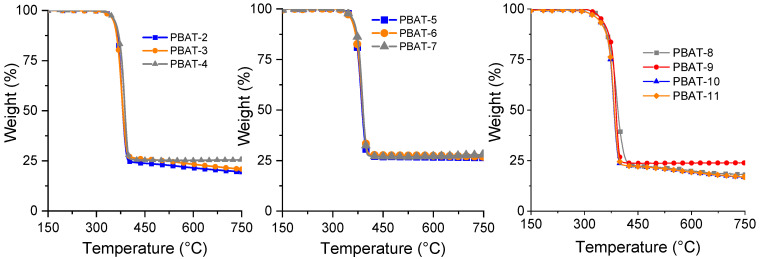
Thermograms of PBAT compositions (N2 atmosphere, 10 °C/min).

**Figure 3 polymers-15-02771-f003:**
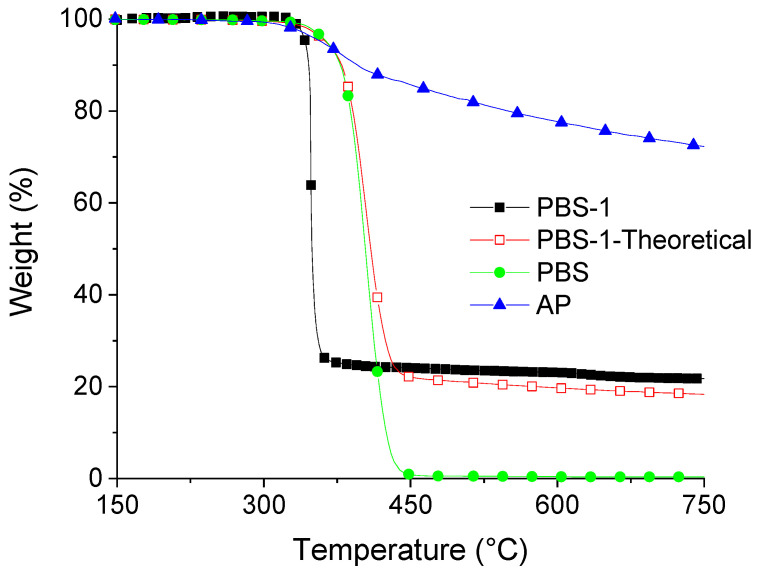
TGA curves (N_2_, 10 °C/min) of pure PBS, AP, and PBS-1 composition (experimental and theoretical).

**Figure 4 polymers-15-02771-f004:**
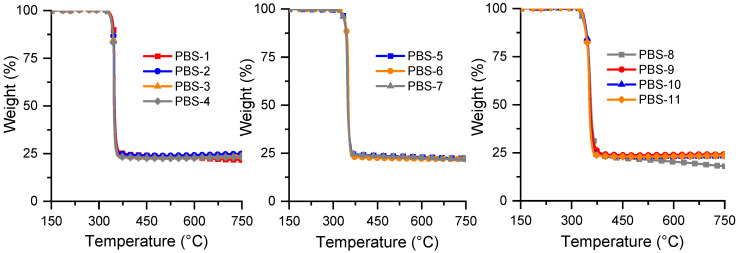
Thermograms of PBS compositions (N2 atmosphere, 10 °C/min).

**Figure 5 polymers-15-02771-f005:**
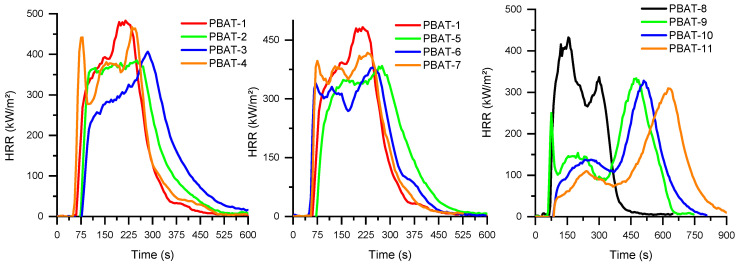
HRR as a function of time for PBAT compositions.

**Figure 6 polymers-15-02771-f006:**
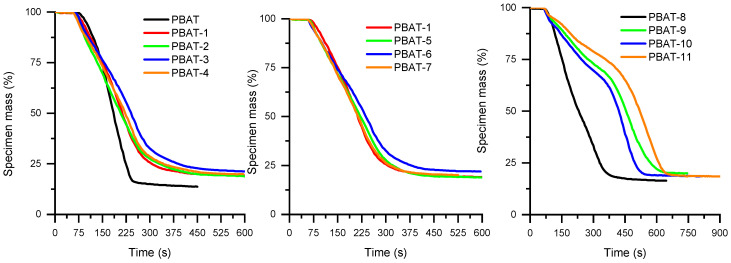
Mass of specimen as a function of time for PBAT compositions.

**Figure 7 polymers-15-02771-f007:**
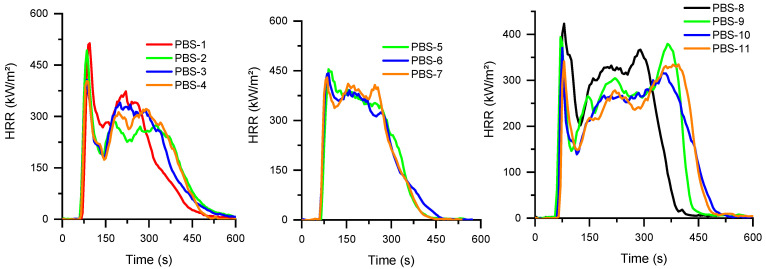
HRR as a function of time for PBS compositions.

**Figure 8 polymers-15-02771-f008:**
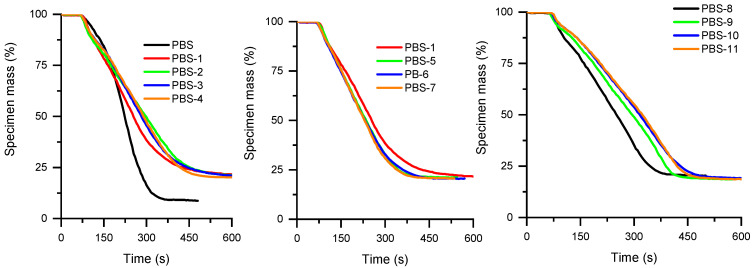
Mass of specimen as a function of time for PBS compositions.

**Figure 9 polymers-15-02771-f009:**
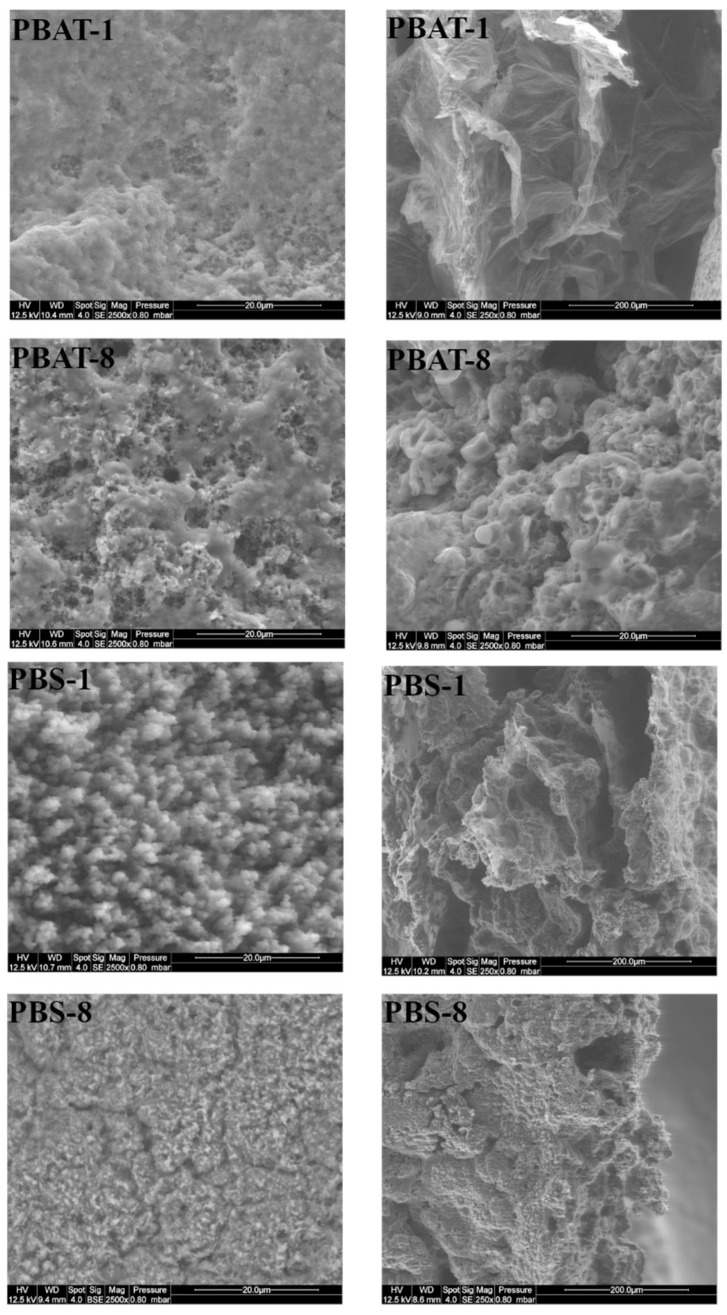
SEM pictures of PBAT and PBS residues containing only flame retardants (left: upper side (× 2500); right: cross-section (× 250)).

**Figure 10 polymers-15-02771-f010:**
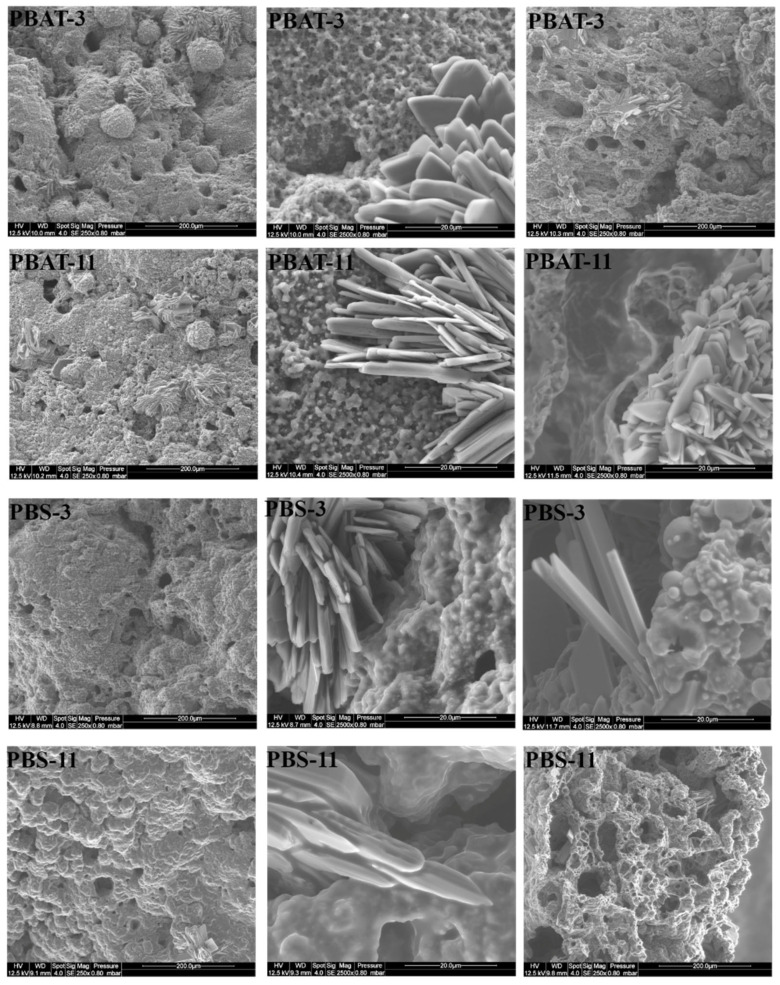
SEM pictures of PBAT and PBS residues containing SuperPozz and APP (PBA-3 and PBS-3), SuperPozz, APP and MPP (PBA-11 and PBS-11), left: upper side (× 250), center: upper side (× 2500), right: cross-section (× 250 or × 2500).

**Figure 11 polymers-15-02771-f011:**
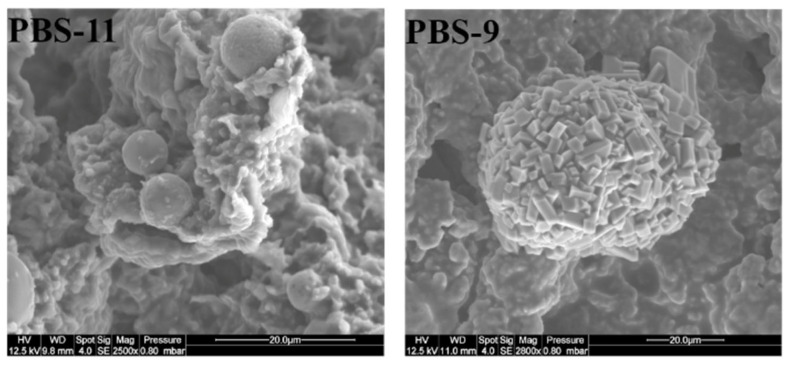
SEM pictures of PBS residues (× 2500) showing the evolution of cenospheres toward new structures.

**Figure 12 polymers-15-02771-f012:**
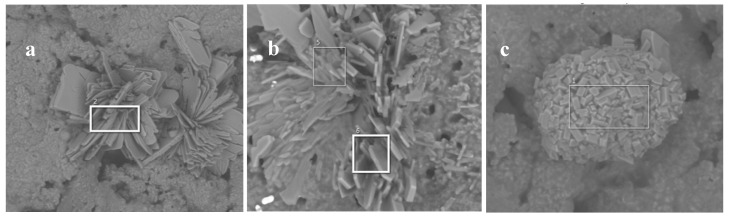
The X-microanalysis of crystallized structure as windroses in PBS-9 (**a**) of the massic composition: P 24.1%, C 13.4%, Al 4.8%, Fe 3.6%, Si 1.0%, Ca 0.3%, and in PBA-9 (**b**) of the massic composition: P 27.0%, C 9.0%, Ca 10.7%, and Al 1.5% as pseudo-spherical aggregates of platelets in PBS-10 (**c**) of the massic composition: P 22.1%, C 14.5%, Ca 8.9% Al 1.0, and Si 0.8%.

**Figure 13 polymers-15-02771-f013:**
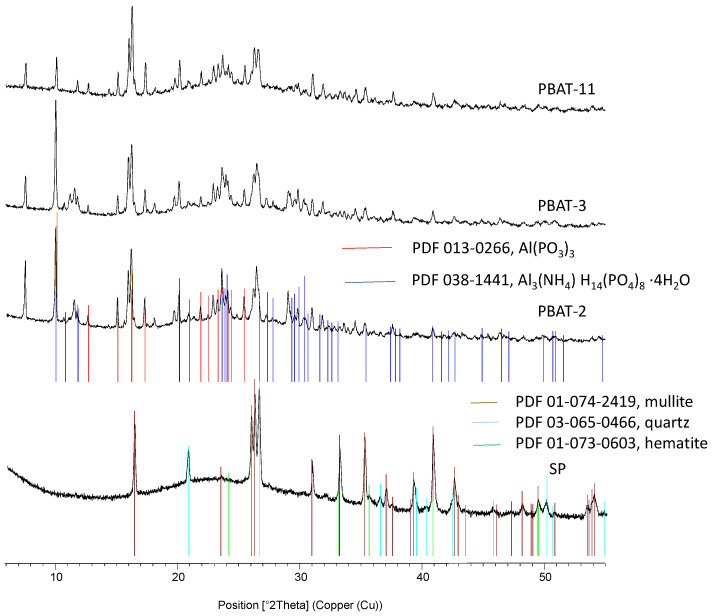
X-ray diffractograms of residues for PBAT compositions and initial fly ash.

**Figure 14 polymers-15-02771-f014:**
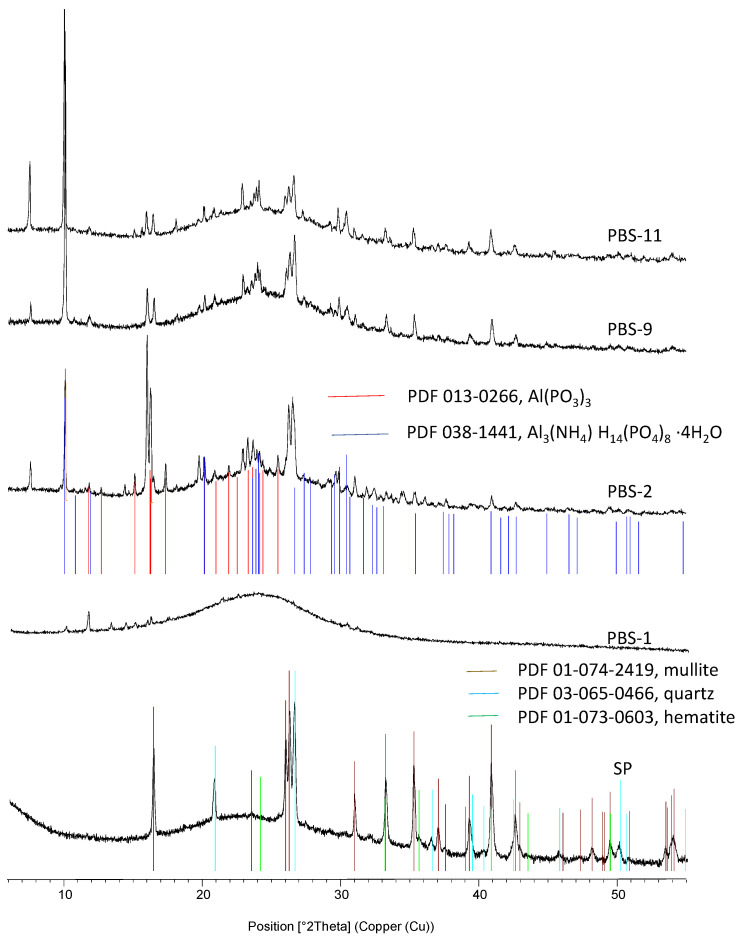
X-ray diffractograms of residues for PBS compositions and initial fly ash.

**Table 1 polymers-15-02771-t001:** Temperature profiles used on twin-screw extrusion of PLA, PBS, and PBAT.

Polymer	Zone 1	Zone 2	Zone 3	Zone 4	Zone 5	Zone 5	Zone 7	Zone 8	Zone 9
PBS	25	110	115	120	125	130	130	130	130
PBAT	60	90	160	180	180	180	180	180	180

**Table 2 polymers-15-02771-t002:** Nomenclature of the prepared PBAT formulations. SP-G corresponds to epoxysilane and SP-T to titanate.

Formulation	PBAT	AP	MP	SP	SP-G	SP-T
PBAT	100					
PBAT-1	75	25				
PBAT-2	75	18.75		6.25		
PBAT-3	75	18.75			6.25	
PBAT-4	75	18.75				6.25
PBAT-5	75	12.5		12.5		
PBAT-6	75	12.5			12.5	
PBAT-7	75	12.5				12.5
PBAT-8	75	16.67	8.33			
PBAT-9	75	12.5	6.25	6.25		
PBAT-10	75	12.5	6.25		6.25	
PBAT-11	75	12.5	6.25			6.25

**Table 3 polymers-15-02771-t003:** Nomenclature of the prepared PBS formulations. SP-A corresponds to aminosilane and SP-T to titanate.

Formulation	PBS	AP	MP	SP	SP-A	SP-T
PBS	100					
PBS-1	75	25				
PBS-2	75	18.75		6.25		
PBS-3	75	18.75			6.25	
PBS-4	75	18.75				6.25
PBS-5	75	12.5		12.5		
PBS-6	75	12.5			12.5	
PBS-7	75	12.5				12.5
PBS-8	75	16.67	8.33			
PBS-9	75	12.5	6.25	6.25		
PBS-10	75	12.5	6.25		6.25	
PBS-11	75	12.5	6.25			6.25

**Table 4 polymers-15-02771-t004:** Thermogravimetric data of PBAT compositions.

Formulation	T_5%_ (°C)	T_20%_ (°C)	T_50%_ (°C)	Residue at 750 °C (%)
PBAT	382	397	410	7
PBAT-1	357	381	393	18
PBAT-2	354	370	382	18.4
PBAT-3	352	368	382	19.4
PBAT-4	356	376	388	25.5
PBAT-5	357	374	388	26.5
PBAT-6	355	375	390	27
PBAT-7	360	378	390	28
PBAT-8	346	374	394	16.9
PBAT-9	347	377	388	23.9
PBAT-10	338	370	383	16.6
PBAT-11	338	370	384	16.8

**Table 5 polymers-15-02771-t005:** Thermogravimetric data of PBS compositions.

Formulation	T_5%_ (°C)	T_20%_ (°C)	T_50%_ (°C)	Residue at 750 °C (%)
PBS	364	389	404	-
PBS-1	342	348	349	22
PBS-2	339	347	348	25
PBS-3	339	346	348	23.6
PBS-4	338	346	347	23.4
PBS-5	337	347	350	21.9
PBS-6	338	349	351	22.1
PBS-7	336	345	349	20.3
PBS-8	332	347	357	16.3
PBS-9	336	348	355	24.2
PBS-10	336	348	353	23.2
PBS-11	335	347	352	23.6

**Table 6 polymers-15-02771-t006:** PCFC data of PBAT compositions.

Formulation	Temperature pHRR (°C)	pHRR (W/g)	THR (KJ/g)	HRC (J/g/°C)	% Residue
PBAT	427	516	23.0	537	1.3
PBAT-1	407	421	19.5	436	11.7
PBAT-2	411	435	18.1	452	16.0
PBAT-3	410	405	18.2	419	18.4
PBAT-4	412	400	18.1	411	19.0
PBAT-5	412	421	18.3	439	19.6
PBAT-6	415	428	18.3	444	21.3
PBAT-7	414	438	18.3	463	21.6
PBAT-8	415	492	17.0	515	14.2
PBAT-9	408	496	18.7	522	17.1
PBAT-10	411	541	18.6	559	16.7
PBAT-11	409	534	18.6	558	16.5

**Table 7 polymers-15-02771-t007:** PCFC data of PBS compositions.

Formulation	Temperature pHRR (°C)	pHRR (W/g)	THR (KJ/g)	HRC (J/g/°C)	% Residue
PBS	425	463	20.7	474	0
PBS-1	385	907	15.8	933	9.2
PBS-2	389	696	16.0	717	19.2
PBS-3	386	716	17.1	742	18.0
PBS-4	388	570	16.6	587	18.7
PBS-5	394	450	16.2	463	20.9
PBS-6	389	654	16.5	672	19.4
PBS-7	391	621	15.9	641	20.7
PBS-8	399	482	15.3	496	15.7
PBS-9	390	874	16.4	908	14.9
PBS-10	392	765	16.7	788	13.9
PBS-11	390	816	16.7	851	15.9

**Table 8 polymers-15-02771-t008:** Cone calorimeter data for PBAT compositions.

Formulation	TTI (s)	pHRR (kW/m^2^)	THR (MJ/m^2^)	MAHRE (kW/m^2^)	Residue (%)	UL94
PBAT	71 ± 3	751 ± 60	93 ± 5	345 ± 50	8 ± 1	NC
PBAT-1	59 ± 2	485 ± 5	91 ± 1	292 ± 6	19 ± 0.1	V-2
PBAT-2	61 ± 2	480 ± 70	94 ± 0.6	282 ± 35	19 ± 0.2	V-2
PBAT-3	64 ± 2	420 ± 20	98 ± 1	235 ± 13	20 ± 0.4	V-2
PBAT-4	57 ± 9	461 ± 10	93 ± 2	283 ± 7	20 ± 0.4	V-2
PBAT-5	59 ± 3	379 ± 8	98 ± 0.3	244 ± 3	21	V-2
PBAT-6	56 ± 6	393 ± 19	91 ± 0.3	262 ± 10	21 ± 0.4	V-2
PBAT-7	56 ± 2	409 ± 11	93 ± 1	280 ± 9	20 ± 0.2	V-2
PBAT-8	66 ± 3	432 ± 4	94 ± 1	246 ± 12	17 ± 0.1	V-2
PBAT-9	55 ± 1	366 ± 38	95 ± 2	159 ± 4	21 ± 0.2	V-2
PBAT-10	61 ± 2	342 ± 20	93±	147 ± 12	21 ± 2	V-2
PBAT-11	60 ± 6	285 ± 28	94 ± 2	120 ± 3	21 ± 2	V-2

**Table 9 polymers-15-02771-t009:** Cone calorimeter data for PBS compositions.

Formulation	TTI (s)	pHRR (kW/m^2^)	THR (MJ/m^2^)	MAHRE (kW/m^2^)	Residue (%)	UL94-V
PBS	71 ± 4	698 ± 22	103 ± 1	314 ± 6	8 ± 0.5	NC
PBS-1	71 ± 2	524 ± 15	92 ± 1	241 ± 2	22 ± 0.5	V-2
PBS-2	63 ± 1	483 ± 15	97 ± 1	225 ± 9	21 ± 0.1	V-2
PBS-3	68 ± 0.7	411 ± 6	97 ± 0.5	228 ± 3	20 ± 0.1	V-2
PBS-4	68 ± 4	410 ± 11	97 ± 1	226 ± 1	20 ± 0.1	V-2
PBS-5	68 ± 5	450 ± 6	96 ± 2	281 ± 8	21 ± 0.1	V-2
PBS-6	67 ± 5	434 ± 11	96 ± 1	272 ± 3	21 ± 0.2	V-2
PBS-7	65 ± 3	425 ± 8	97 ± 0.5	282 ± 5	20 ± 0.1	V-2
PBS-8	60 ± 1	419 ± 6	92 ± 1	307 ± 31	20 ± 0.3	V-0
PBS-9	61 ± 6	383 ± 16	98 ± 1	225 ± 15	19 ± 0.2	V-0
PBS-10	63 ± 2	375 ± 6	97±	214 ± 4	19 ± 0.1	V-0
PBS-11	68 ± 2	338 ± 4	98 ± 1	213 ± 3	19 ± 0.1	V-0

**Table 10 polymers-15-02771-t010:** EDX results of selected cone calorimeter residues of PBS formulations.

Formulation	Ca (wt.%)	Al (wt.%)	Si (wt.%)	Fe (wt.%)	C (wt.%)	P (wt.%)	N (wt.%)	Phosphorus Balance (%)
PBS-1	-	-	-	-	34	24	4	65
PBS-3	1	4	6	1	12	25	3	85
PBS-8	-	-	-	-	23	28	4	
PBS-9	1	5	7	1	16	21	-	78
PBS-10	1	4	6	0.5	17	21	-	81
PBS-11	1	4	6	0.5	16	22	-	84

**Table 11 polymers-15-02771-t011:** EDX results of selected cone calorimeter residues of PBAT formulations.

Formulation	Ca (wt.%)	Al (wt.%)	Si (wt.%)	Fe (wt.%)	C (wt.%)	P (wt.%)	N (wt.%)	Phosphorus Balance (%)
PBAT-1	-	3	-	-	25	26	2	53
PBAT-3	0.5	5	8	0.5	11	23	-	82
PBAT-8	-	0.5	0.2	-	8	29	4	
PBAT-9	0.5	5	5	1	12	25	-	100
PBAT-10	1	5	6	0.5	8	26	-	100
PBAT-11	1	5	5	1	10	26	-	100

## Data Availability

Data available on request due to restrictions, e.g., privacy or ethical.

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
