# Peer review of "Incorporation of Fly Ash in Flame-Retardant Systems of Biopolyesters"

_polymers, 2023, doi:10.3390/polym15132771_

Round 1

Reviewer 1 Report

In this study, fly ash was added as synergistic flame retardant together with ammonium polyphosphate and/or melamine polyphosphate for Poly Butyl Succinate (PBS) and Poly Butyl Adipate Tereph-talate (PBAT). Many analyses were carried out and good results were also obtained. There are some questions for better clarification:

1.      The whole results section is difficult to understand due to the large number of samples and lack of effective comparison.

2.      Both PCFC and cone calorimeter evaluate the heat release capacity of the samples. Is there any difference between the results of these two tests?

3.      How about the smoke release performance of the samples?

4.      Why do the residual chars after combustion contain metal elements? Will they influence the flame retardancy of the samples?

5.      Please inform all chemicals used.

6.      Abstract – it is difficult to analyses the phosphorus compounds by using XRD.

7.      Please highlights the role of fly ash in the flame-retardant systems which are complex.

In this study, fly ash was added as synergistic flame retardant together with ammonium polyphosphate and/or melamine polyphosphate for Poly Butyl Succinate (PBS) and Poly Butyl Adipate Tereph-talate (PBAT). Many analyses were carried out and good results were also obtained. There are some questions for better clarification:

1.      The whole results section is difficult to understand due to the large number of samples and lack of effective comparison.

2.      Both PCFC and cone calorimeter evaluate the heat release capacity of the samples. Is there any difference between the results of these two tests?

3.      How about the smoke release performance of the samples?

4.      Why do the residual chars after combustion contain metal elements? Will they influence the flame retardancy of the samples?

5.      Please inform all chemicals used.

6.      Abstract – it is difficult to analyses the phosphorus compounds by using XRD.

7.      Please highlights the role of fly ash in the flame-retardant systems which are complex.

Author Response

In this study, fly ash was added as synergistic flame retardant together with ammonium polyphosphate and/or melamine polyphosphate for Poly Butyl Succinate (PBS) and Poly Butyl Adipate Tereph-talate (PBAT). Many analyses were carried out and good results were also obtained. There are some questions for better clarification:

Dear reviewer, thank you for your suggestions. Please find below some comments and proposed modifications on our manuscript.

  1. The whole results section is difficult to understand due to the large number of samples and lack of effective comparison.

We agree that the number of samples is large, but we tried to be as clear as possible.

  1. Both PCFC and cone calorimeter evaluate the heat release capacity of the samples. Is there any difference between the results of these two tests?

Similarly to cone calorimetry, PCFC calculates the heat release rate by measuring the consumption of oxygen, according to the Huggett’s relation. In PCFC, a few milligrams sample is pyrolyzed under nitrogen flow according to a heating ramp (typically 1K/s) up to 750°C. The gases released during the pyrolysis are evacuated into an oven at 900°C in the presence of a 80/20 N2/O2 mixture. In these conditions, a total combustion of these gases takes place. Correlations between the cone calorimeter and the PCFC results is not general, for three main reasons. The first is the barrier effect, which is not efficient in the PCFC, in contrast to the cone calorimeter. The barrier effect becomes effective when an insulating layer could protect the underlying material from the heat source. This layer may be composed of char and/or mineral particles and should be thermally stable. The second is related to thermal stability. Two polymers could exhibit similar peaks of heat release rate but different degradation temperatures in PCFC. In this case, cone calorimeter results could be very different. The third reason is that the combustion is complete in the PCFC standard test while the combustion efficiency could be less than 1 in the cone calorimeter test, even if this test is carried out under well-ventilated conditions [1].

  1. 1. Lin, T.S.; Cogen, J.; Lyon, R. Correlations between Microscale Combustion Calorimetry and Conventional Flammability Tests for Flame Retardant Wire and Cable Compounds. In Proceedings of the 56th International Wire & Cable Symposium; 2007.

  1. How about the smoke release performance of the samples?

The addition of APP to PBAT or PBS leads to an increase on Total Smoke Release (TSR) compared to neat polymer. The partial replacement of APP by fly ash does not lead to significant differences. Furthermore, the addition of melamine polyphosphate (MP) leads to a small decrease on TSR.

  1. Why do the residual chars after combustion contain metal elements? Will they influence the flame retardancy of the samples?

Metal elements are present on fly ash and thus remains on residue at the end of cone calorimeter test. These elements may have a limited influence on fire behaviour due to its limited amount.

  1. Please inform all chemicals used.

All chemicals used were informed.

  1. Abstract – it is difficult to analyses the phosphorus compounds by using XRD.

Various works on literature showed the presence of new phosphorus species due to an interaction between constituents, in particular with the presence of inorganic particles as clays. In a previous work on a FR system using fly ash and ammonium polyphosphate, using XRD and NMR we highlighted the formation of new phosphate compounds.

  1. Please highlights the role of fly ash in the flame-retardant systems which are complex.

Fly ash behaves as reactive component with both flame retardants, leading to the formation of new crystalline and amorphous compounds. Synergistic effects on fire performance resulting from the reactive behaviour of fly ash can be highlighted and are ascribed to fire retardant mechanisms acting in the condensed phase, and corresponding to the conservation of high phosphorus contents in the condensed phase.

Reviewer 2 Report

In this work, the incorporation of fly ash in PBS and PBAT as component of flame retardant systems has been studied. It is interesting with some valuable results. However, there are still some issues that should be addressed and more experiments and discussion should be supplied before further consideration.

1. In introduction, the following article reporting flame retardant PLA composites by using phytate may be useful to improve the literature review: Composites Part A: Applied Science and Manufacturing 2018, 110, 227-236.

2. In abstract and section 2.2.3, the authors claimed that UL94V was used to evaluate the fire reaction of the composites. However, there are no experimental results in this manuscript.

3. In section 3.2, the authors claimed that all compositions lead to a decrease of pHRR. However, the pHRR values of major PBS composites are higher than that of pure PBS (Table 7). For example, the loading of 25 wt% AP leads to the increase of pHRR from 463 W/g for neat PBS to 907 W/g for PBS-1. Why does this phenomenon appear? Please clarify.

4. In Table 8, it can be seen that the addition of flame retardants into PBAT leads to the reduction of pHRR. However, the THR of PBAT composites is slightly influenced. Why? Please clarify.

5. Can the authors supplement tensile tests of PBAT and PBS composites to evaluate their mechanical properties?

Author Response

In this work, the incorporation of fly ash in PBS and PBAT as component of flame retardant systems has been studied. It is interesting with some valuable results. However, there are still some issues that should be addressed and more experiments and discussion should be supplied before further consideration.

Dear reviewer, thank you for your suggestions. Please find below some comments and proposed modifications on our manuscript.

  1. In introduction, the following article reporting flame retardant PLA composites by using phytate may be useful to improve the literature review: Composites Part A: Applied Science and Manufacturing 2018, 110, 227-236.

Thank you for suggestion. We changed the introduction accordingly.

  1. In abstract and section 2.2.3, the authors claimed that UL94V was used to evaluate the fire reaction of the composites. However, there are no experimental results in this manuscript.

Sorry for this mistake. We added the results of UL-94 tests on tables 8 and 9. The text was also changed accordingly.

  1. In section 3.2, the authors claimed that all compositions lead to a decrease of pHRR. However, the pHRR values of major PBS composites are higher than that of pure PBS (Table 7). For example, the loading of 25 wt% AP leads to the increase of pHRR from 463 W/g for neat PBS to 907 W/g for PBS-1. Why does this phenomenon appear? Please clarify.

This behavior can be related to APP decomposition at low temperature when it is added to PBS. Since APP decomposes by releasing water, an early degradation of PBS due to hot hydrolysis may be observed. This behavior was also observed in literature for PBS/APP/flax with and increase on pHRR on PCFC and a decrease on TTI on cone calorimeter.

  1. In Table 8, it can be seen that the addition of flame retardants into PBAT leads to the reduction of pHRR. However, the THR of PBAT composites is slightly influenced. Why? Please clarify.

A decrease on pHRR may not lead to a decrease on THR. Total Heat Release is a measure of the amount of heat energy evolved during the burning time and it is calculated as the area under the HRR x time curve. When comparing the HRR curves of compositions, a reduction on pHRR can be observed accompanied by a longer combustion time, implying small changes on total heat released on the formulations evaluated on ours work.

  1. Can the authors supplement tensile tests of PBAT and PBS composites to evaluate their mechanical properties?

Mechanical properties of some PBAT and PBS samples were added. Please refer to Supplementary Information.

Reviewer 3 Report

The paper from Batistella et al. investigates the fire behavior of PBS and PBAT in the presence of fly ash with ammonium polyphosphate and/or melamine polyphosphate (total loading: 25 wt.%); besides, the fly ash is surface treated with two silanes and a titanate. The manuscript shows some novelty and quite rational development, although the conclusions could be be better supported by the experimental data.

In particular:

- TG analyses should also be performed in air 8at least on the most representative formulations) and the obtained results commented

- the modification of the surface of fly ash should be better assessed: to this aim, FTIR-ATR spectroscopy could be exploited

- what about the smoke parameters in cone calorimeter tests?

- usually, the incorporation of inorganic filler into polymer matrices makes the latter stiffer and more brittle: it should be reasonable to assess the effect of the filler (at least for the most representative formulations) on the mechanical behavior of the FR compounds (tensile or flexural tests should be performed)

- synergism could be better assessed in a quantitative way, by measuring the synergistic effectiveness parameter, as nicely depicted by Horrocks or by Lewin.

Author Response

The paper from Batistella et al. investigates the fire behavior of PBS and PBAT in the presence of fly ash with ammonium polyphosphate and/or melamine polyphosphate (total loading: 25 wt.%); besides, the fly ash is surface treated with two silanes and a titanate. The manuscript shows some novelty and quite rational development, although the conclusions could be be better supported by the experimental data.

Dear reviewer, thank you for your suggestions. Please find below some comments and proposed modifications on our manuscript.

In particular:

- TG analyses should also be performed in air 8at least on the most representative formulations) and the obtained results commented

We agree that TG analyses under air is important to understand the thermo-oxidation of PBS and PBAT. Moreover, our TGA equipment is out of service and we are not able to conduct TGA tests.

- the modification of the surface of fly ash should be better assessed: to this aim, FTIR-ATR spectroscopy could be exploited

FTIR-ATR test may give some information on the surface modification of fly ash. Moreover, as the coupling agent amount is very low (1wt.%) it was not possible to observe differences on FTIR spectra. Moreover, we conducted a Turbiscan™ equipment, which allowed us to access the differences on surface modifications.

- what about the smoke parameters in cone calorimeter tests?

The addition of APP to PBAT or PBS leads to an increase on Total Smoke Release (TSR) compared to neat polymer. The partial replacement of APP by fly ash does not lead to significant differences compared to the formulation containing only APP. Furthermore, the addition of melamine polyphosphate (MP) leads to a small decrease on TSR.

- usually, the incorporation of inorganic filler into polymer matrices makes the latter stiffer and more brittle: it should be reasonable to assess the effect of the filler (at least for the most representative formulations) on the mechanical behavior of the FR compounds (tensile or flexural tests should be performed)

Mechanical properties of some PBAT and PBS samples were added. Please refer to Supplementary Information.

- synergism could be better assessed in a quantitative way, by measuring the synergistic effectiveness parameter, as nicely depicted by Horrocks or by Lewin.

We agree that effectiveness parameter described by Lewin or Horrocks could give some insights about the synergistic effects between components. Unfortunately, we are do not have all the required formulations to apply the effectiveness parameter and due to the restricted time to submit the reviewed version we will not be able to prepare it.

Round 2

Reviewer 1 Report

The manuscript can be accepted for publication.

Author Response

Dear Reviewer. Thank you for your time and valuable comments.

Reviewer 2 Report

This manuscript has been improved by the authors. It can be accepted for publication.

Author Response

(The authors gave the same response as above.)

Reviewer 3 Report

The authors have improved the overall quality of the manuscript. Therefore, it can be accepted in its present form.

Author Response

(The authors gave the same response as above.)
